# Determination of rSpike Protein by Specific Antibodies with Screen-Printed Carbon Electrode Modified by Electrodeposited Gold Nanostructures

**DOI:** 10.3390/bios12080593

**Published:** 2022-08-03

**Authors:** Maryia Drobysh, Viktorija Liustrovaite, Ausra Baradoke, Roman Viter, Chien-Fu Chen, Arunas Ramanavicius, Almira Ramanaviciene

**Affiliations:** 1State Research Institute Center for Physical and Technological Sciences, Sauletekio Ave. 3, LT-10257 Vilnius, Lithuania; maryia.drobysh@ftmc.lt (M.D.); ausra.baradoke@ftmc.lt (A.B.); 2NanoTechnas—Center of Nanotechnology and Materials Science, Faculty of Chemistry and Geosciences, Vilnius University, Naugarduko Str. 24, LT-03225 Vilnius, Lithuania; viktorija.liustrovaite@chgf.stud.vu.lt (V.L.); almira.ramanaviciene@chf.vu.lt (A.R.); 3Institute of Atomic Physics and Spectroscopy, University of Latvia, Jelgavas Street 3, LV-1004 Riga, Latvia; roman.viter@lu.lv; 4Center for Collective Use of Research Equipment, Sumy State University, 31 Sanatorna Street, 40000 Sumy, Ukraine; 5Institute of Applied Mechanics, National Taiwan University, 1 Sec. 4, Roosevelt Rd., Da’an Dist., Taipei City 106, Taiwan; stevechen@iam.ntu.edu.tw

**Keywords:** COVID-19, SARS-CoV-2 virus, electrochemical immunosensor, differential pulse voltammetry (DPV), cyclic voltammetry (CV), electrochemical impedance spectroscopy (EIS), self-assembled monolayer (SAM), antigen–antibody complex, spike protein (Spike), gold nanostructures (AuNS)

## Abstract

In this research, we assessed the applicability of electrochemical sensing techniques for detecting specific antibodies against severe acute respiratory syndrome coronavirus 2 (SARS-CoV-2) spike proteins in the blood serum of patient samples following coronavirus disease 2019 (COVID-19). Herein, screen-printed carbon electrodes (SPCE) with electrodeposited gold nanostructures (AuNS) were modified with L-Cysteine for further covalent immobilization of recombinant SARS-CoV-2 spike proteins (rSpike). The affinity interactions of the rSpike protein with specific antibodies against this protein (anti-rSpike) were assessed using cyclic voltammetry (CV) and differential pulse voltammetry (DPV) methods. It was revealed that the SPCE electroactive surface area increased from 1.49 ± 0.02 cm^2^ to 1.82 ± 0.01 cm^2^ when AuNS were electrodeposited, and the value of the heterogeneous electron transfer rate constant (*k*^0^) changed from 6.30 × 10^−5^ to 14.56 × 10^−5^. The performance of the developed electrochemical immunosensor was evaluated by calculating the limit of detection and limit of quantification, giving values of 0.27 nM and 0.81 nM for CV and 0.14 nM and 0.42 nM for DPV. Furthermore, a specificity test was performed with a solution of antibodies against bovine serum albumin as the control aliquot, which was used to assess nonspecific binding, and this evaluation revealed that the developed rSpike-based sensor exhibits low nonspecific binding towards anti-rSpike antibodies.

## 1. Introduction

Severe acute respiratory syndrome coronavirus 2 (SARS-CoV-2) is a highly transmissible and pathogenic coronavirus that first appeared in late 2019 and has since created a pandemic of acute respiratory sickness known as ’coronavirus disease 2019’ (COVID-19), which poses a threat to human health since human-to-human transmission has grown significantly [1]. The progression of the COVID-19 pandemic has shown that there is a crucial need to develop quick and accurate tests to better control the spread of the disease and monitor illness progression. Immunosensors are the most suitable type of sensors for this purpose since they can be used to detect the SARS-CoV-2 virus, confirming the presence of the disease in the individual, or to monitor antibodies against the virus to check for past illness or immunity [2].

The application of electrochemical immunosensors [3,4] and other affinity sensors [5,6] has additional advantages such as cost-effectiveness, ease of use, point-of-care detection, and reduced sample analysis time, which can significantly help in the early diagnosis of COVID-19 disease [7]. The electrochemical transducer element may be directly coupled to an electrochemical biosensor to provide analytical information about the target species via the biochemical or chemical receptor [8]. Several novel systems for the detection of SARS-CoV-2 spike protein have been proposed. Although most of the sensors have been proven to be successful in detecting SARS-CoV-2, several of them lack sensitivity and/or selectivity, have a low sampling rate, and are designed to use a complicated electrode manufacturing technique [9]. In this regard, nanoparticles may offer a viable solution to the sensitivity and selectivity issues [10,11]. Furthermore, some qualitative SARS-CoV-2 antibodies against spike protein methods have been created, allowing for confirmation of antibodies present in the blood; however, immunosensors for quantitative detection of antibodies have been less reported [12].

Carbon is an excellent platform for antibody immobilization features such as large surface area, good conductivity, and high stability [13]. Nonetheless, covalently attaching biomolecules to carbon remains difficult, whereas physisorption does not typically generate permanent coatings and does not allow for control of antibody orientation. The linkage of biomolecules through self-assembled monolayers (SAMs) on a gold substrate has been widely reported in the biosensorics-related literature [14]. However, gold electrodes (or gold screen-printed electrodes (SPCE)) have scarcely been used, due to high cost of gold [15]. Nanostructured metals such as Au nanostructures (AuNS) result in stronger, more defined binding, e.g., amine/carboxy terminated alkanethiols for N-hydroxysuccinimide/N-(3-dimethylaminopropyl)-N’-ethyl-carbodiimide hydrochloride (NHS/EDC) coupling. Furthermore, AuNS can boost the rate of heterogeneous electron transfer, resulting in increased detection sensitivity [16].

For the development of electrochemical immunosensors for the determination of proteins, the surface of the working electrode must be modified by the protein-recognizing antibodies, receptors, or some artificial structures [5,9]. A SAM is often employed for electrode surface modification purposes, and SAMs terminated by –COOH groups are the most suitable for selective and stable rSpike protein immobilization [17]. In this work, L-Cysteine was used for the presence of functional groups such as thiol (–SH), which has a high affinity towards metallic gold and attaches strongly to the gold surface due to the gold–sulphur interaction. This is useful for well-oriented protein immobilization on gold-based transducers [18].

Total internal reflection ellipsometry [7], scanning electrochemical microscopy, surface plasmon resonance [19], quartz crystal microbalance methods, colorimetry [20], electrochemiluminescence [21,22,23,24], electrochemical techniques [25], and other methods [26] are among those that can be used to determine analytical signals generated by affinity sensors. When using the techniques of differential pulse voltammetry (DPV) and cyclic voltammetry (CV), the current response is proportional to the analyte concentration [27]. In our previous work [21], the affinity interaction of recombinant spike protein (rSpike) with antibodies against rSpike (anti-rSpike) was detected using two electrochemical methods: CV and electrochemical impedance spectroscopy (EIS). EIS is frequently used to analyse films produced on electrodes because the EIS method is able to discriminate between various conductivity-/capacitance-related processes that occur on the electrode/solution interface. EIS results are frequently assessed using the corresponding electrical circuit, in which factors such as electrolyte resistance, ionic conductivity, electrical double-layer capacitance, and electron transfer resistance may be distinguished and calculated. Regardless of the fact that EIS is rarely used for analytical purposes, the redox process or analyte-related CV features can be utilized for quantitative findings; however, due to its limitations, EIS is more typically used for exploratory purposes such as assessing the redox process for diverse analytes [28]. In general, pulse techniques such as DPV are more sensitive than linear-sweep-based methods, since CV is the technique most frequently employed for exploratory purposes. Thus, it is rather common in sensor development to employ both these techniques, because CV provides critical information on aspects such as process reversibility and the types of redox processes occurring during the analysis at the interface between the electrode and the solution, whereas potential-pulse-based techniques sometimes enable simplification of the quantification of the analyte [29]. The miniaturization of electrochemical systems enables the determination of protein-based analytes in rather small volumes of aliquots [30].

Therefore, in our present work we compare the applicability of both these two voltametric sensing methods (namely, DPV and CV), taking into account the advantages of their durability and low detection limits in small volumes of aliquots.

To achieve this goal, a label-free electrochemical immunosensor based on SPCE modified with AuNS (SPCE/AuNS) and rSpike protein was designed. The functionalization of SPCE by rSpike was accomplished by the formation of a SAM, L-cysteine (SPCE/AuNS/SAM/rSpike). The affinity reaction was monitored by measuring the decrease in the DPV and CV responses of an [Fe(CN)_6_]^3−/4−^ redox probe recorded upon the addition of an anti-rSpike-containing sample (SPCE/AuNS/SAM/rSpike/anti-rSpike). The created immunoplatform met the sensitivity, selectivity, and repeatability criteria and was successfully used to detect anti-rSpike.

## 2. Experimental

### 2.1. Materials

Tetrachloroauric acid trihydrate (HAuCl_4_·3H_2_O) (99%, CAS# 16961-25-4), KNO_3_ (≥99.0%, CAS# 7757-79-1), ethanol (EtOH) (99.9%, CAS# 64-17-5), L-Cysteine (97%, CAS# 52-90-4), N-(3-dimethylaminopropyl)-N’-ethyl-carbodiimide hydrochloride (EDC) (≥99.0%, CAS# 25952-53-8), ethanolamine (EA) (≥98%, CAS# 141-43-5), K_3_Fe(CN)_6_ (≥99.0%, CAS# 13746-66-2), K_4_Fe(CN)_6_ (≥99.0%, CAS# 14459-95-1), and phosphate-buffered saline (PBS) tablets, pH 7.4, were obtained from Sigma–Aldrich (Steinheim, Germany). N-hydroxysuccinimide (NHS) (98.0%, CAS# 6066-82-6) was purchased from Alfa Aesar (Karlsruhe, Germany). Anti-bovine albumin (BSA) antibodies (anti-BSA) were obtained from Biotecha, Lithuania. All reagents were analytical grade and were used without additional purification. All aqueous solutions were prepared in deionized water.

The SARS-CoV-2 recombinant spike protein of SARS-CoV-2 (rSpike) was produced by Baltymas (Vilnius, Lithuania) [31]. Serum samples containing antibodies (anti-rSpike) of volunteers vaccinated with a single dose of the Vaxzevria vaccine who had COVID-19 after two weeks were collected [10] according to Lithuanian ethics law. The ethics committee’s permission was not required for this project (as confirmed by the Vilnius Regional Biomedical Research Ethics Committee).

### 2.2. Electrochemical Measurements

Electrochemical characterization of the working surface was performed using a potentiostat controlled by the DStat interface software from Wheeler Microfluidics Lab (University of Toronto, Toronto, ON, Canada). DRP-110 screen-printed carbon electrode systems (SPCEs), which are based on a working electrode (geometric area of 0.126 cm^2^), a carbon counter, and Ag/AgCl reference electrodes, were purchased from Metrohm DropSens (Oviedo, Spain). SPCEs were connected through a specialized ‘box-connector’ for screen-printed electrodes (DRP-DSC, DropSens, Oviedo, Spain).

All electrochemical measurements were performed in 0.1 M PBS, pH 7.4 solution, adding 2 mM K_3_Fe(CN)_6_/K_4_Fe(CN)_6_ ([Fe(CN_6_)]^3−/4−^) solution as a redox probe. Electrochemical characterization of the working electrode at different modification stages was carried out using DPV and CV. DPV experiments were measured in the potential range from −0.4 to +0.6 V vs. Ag/AgCl, with a step size of 0.004 V. CV was registered in the potential window from −0.4 to +0.6 V vs. Ag/AgCl, at a scan rate of 0.05 V/s. All experiments were performed at room temperature (20 °C).

Scanning electron microscope (SEM) images were acquired with a scanning electron microscope (Hitachi-70 S3400 N VP-SEM). 

### 2.3. Au Deposition on SPCE

The SPCE was covered with 100 µL of the solution containing 0.1 M KNO_3_ and 5 mM HAuCl_4_. Electrodeposition was performed at a potential of -0.4 V for 60 s. Then, after AuNS deposition on the SPCE (SPCE/AuNS), the electrode was rinsed with deionized water and dried under a N_2_ (%) flow (Figure 1, step 1).

### 2.4. Immobilisation of rSpike and Anti-rSpike

The SPCE/AuNS were incubated at 20 °C for 4 h in 5 mM L-Cysteine ethanolic solution to form a self-assembled monolayer (SAM) on the working surface (SPCE/AuNS/SAM) (Figure 1, step 2). After incubation, the SPCE/AuNS/SAM electrode was rinsed with deionized water and then dried under a N_2_ flow. SPCE/AuNS/SAM was activated with 10 µL of a mixture of 0.02 M EDC and 0.005 M NHS in PBS, pH 7.4, for 10 min. After the activation, the electrode was incubated with 10 µL of 50 µg/mL rSpike in PBS, pH 7.4, at 20 °C for 20 min. Immobilization of rSpike was performed through covalent coupling of the protein’s primary amine functional groups and the activated carboxylic groups of the SAM (SPCE/AuNS/SAM/rSpike) (Figure 1, step 3). The remaining reactive esters were deactivated by incubating with a 1 mM water solution of ethanolamine for 10 min. Afterwards, SPCE/AuNS/SAM/rSpike was incubated with 10 µL of anti-rSpike in PBS, pH 7.4, with a concentration range from 0.5 to 3.5 nM, at 20 °C for 10 min (SPCE/AuNS/SAM/rSpike/anti-rSpike) (Figure 1, step 4). After each stage of incubation, the system was rinsed with deionized water and used for further electrochemical measurements.

### 2.5. Calibration of Anti-rSpike

The initial number of binding antibody units (BAU) per mL against the spike protein of SARS-CoV-2 in the serum sample was 5860 BAU/mL. The concentration was defined by a chemiluminescent microparticle immunoassay performed in the laboratory of Tavo Klinika, Ltd. (Vilnius, Lithuania). The target antibodies in the sample were recalculated from BAU/mL to nM using the ratio 1 BAU/mL: 20 ng/mL (considering the molecular weight of immunoglobulin G as ~150 kDa) [32,33,34].

Calibration curves were obtained by the incubation of SPCE/AuNS/SAM/rSpike in serum samples containing 0.5, 1.0, 1.5, 2.5, and 3.5 nM of anti-rSpike, for 10 min for each concentration. DPV and CV data were used to plot the calibration curves. The relative response (RR%) used for the evaluation of the method specificity was calculated using the equation RR% = ((X_i_ − µX_blank_)/(X_blank_)) × 100%, where X_i_ is the response for concentration i and X_blank_ is the response for a blank. 

## 3. Results and Discussion

### 3.1. Electrochemical Characterisation of SPCE and SPCE/AuNS

In order to improve the surface area for rSpike immobilization and to facilitate better electron transfer kinetics, electrochemical deposition of AuNS was performed on the SPCE working electrode. The CV and DPV results are provided in Figure 2. In addition, the electroactive surface area for SPCE/AuNS was determined using CV in 10 mM H_2_SO_4_ (Figure 3). The characteristic gold reduction and oxidation peaks are present in the potential window from 0 to +1.0 V [35], while the measurements for the unmodified SPCE surface reveal no oxidation or reduction peaks (Figure 3, inset).

With the aim of evaluating the electrochemical performance of the sensor, it is critical to quantify the electrochemically active surface area of the substrate material [36], as well as to define the heterogeneous electron transfer rate constant (*k*^0^) [37]. For this purpose, CV at a range of scan rates from 0.01 to 0.15 V/s was performed in PBS, pH 7.4, containing 2 mM [Fe(CN_6_)]^3−/4−^ for both SPCE and SPCE/AuNS (Figure 4, Table 1). 

Using the Randles–Sevcik equation, the electrochemically active surface areas were calculated as 1.49 ± 0.02 cm^2^ for SPCE and 1.82 ± 0.01 cm^2^ for SPCE/AuNS (Figure 5A). The difference between the values can be explained by the increase in the surface roughness (Figure 6), thus improving the working substrate properties for the subsequent immobilization of the biorecognition element. Furthermore, the data obtained from CV at different scan rates allowed us to assess *k*^0^ by means of the improved Nicholson’s approach for the quasi-reversible electrochemical reaction [38,39]. The value for SPCE was 6.30 ± 0.13 × 10^−5^, while that for SPCE/AuNS was 14.56 ± 0.20 × 10^−5^ (Figure 5B), which is more than twice as high. Thus, it can be concluded that the electrodeposition of AuNS contributes not only to an increase in the electrode active area but also to the rate of heterogeneous electron transfer.

For further investigations of the electrochemical surface properties, CV and DPV measurements in 10 mM PBS, pH 7.4, containing 2 mM [Fe(CN_6_)]^3−/4−^ were performed for SPCE and SPCE/AuNS in the potential range from −0.4 to +0.6 V (Figure 2).

DPV is known to be a potentiostatic method, suggesting some advantages over conventional methods such as CV. In the waveform, DPV is a series of pulses, while for CV the potential is ramped linearly with time. Due to the minimization of the capacitive current, pulse methods, including DPV, are considered to be more sensitive than linear sweep methods. On the other hand, CV is the method most frequently used for research purposes. Hence, it is quite a common practice in sensor development to use both types of electrochemical methods. While CV reveals key electrochemical characteristics such as process reversibility and reflects the redox processes that occur in the system, DPV is employed for quantitative analysis [40].

Since the obtained cyclic voltammograms were quasi-reversible [41], the character of the correlation between the current peak intensity and the surface modification step was not the same for cathodic and anodic peaks. For instance, in Figure 2, the resolution of the current density signals in the anodic region was higher than in cathodic region. This trend increased with further surface modification, leading to the overlapping of the cathodic peaks (Figure 7). Hence, to facilitate quantitative data analysis, we used the values of the anodic current density (*j*_pa_) as the analytical parameter gained from the CV experiments.

As shown in Figure 2, cyclic and differential pulse voltammograms revealed the same trend of increasing current densities after the working surface modification. Specifically, the values increased from 394.71 ± 0.69 to 536.30 ± 0.42 and from 274.89 ± 0.17 to 632.53 ± 0.83 µA/cm^2^ for CV and DPV, respectively. Potential values were also changed, moving left along the axis. This indicates a substrate material change with increasing the conductivity.

### 3.2. Electrochemical Characterisation of the Biosensing Element

CV and DPV in 10 mM PBS, pH 7.4, with 2 mM [Fe(CN_6_)]^3−/4−^ as a redox probe were performed and evaluated for SPCE/AuNS, SPCE/AuNS/SAM, and SPCE/AuNS/SAM/rSpike (Figure 7, Table 2). The CV oxidation peaks were compared after each of the above-mentioned stages of the biosensing element formation.

As considered in the previous section, CV for SPCE/AuNS was characterized by a voltammogram with sharp oxidative/reductive peaks and with a *j*_pa_ value of 536.30 ± 0.42 µA/cm^2^. After SPCE/AuNS/SAM formation, a decrease in *j*_pa_ to 436.96 ± 0.18 µA/cm^2^ was observed. Then, the activation of the terminal –COOH group of the L-Cysteine took place without accompanying electrochemical measurements, to ensure subsequent effective rSpike immobilization. Afterwards, the remainder of the activated functional groups of the SAM were blocked by 1 mM ethanol amine, to avoid nonspecific interactions during the anti-rSpike coupling stages. CV after antigen immobilization with SPCE/AuNS/SAM/rSpike formation and blocking revealed a further current density decrease to 361.83 ± 0.28 µA/cm^2^.

DPV measurements for the above-mentioned stages of biosensing element formation showed the same tendency toward a stepwise decrease in the current density to 632.53 ± 0.83, 363.52 ± 0.28, and 185.26 ± 1.17 µA/cm^2^ for SPCE/AuNS, SPCE/AuNS/SAM, and SPCE/AuNS/SAM/rSpike. These results are summarized in Table 1.

The decrease in current density according to both CV and DPV methods can be explained by the increasing layer thickness on the working electrode surface, thus hampering electron transfer. The stepwise broadening of the DPV peaks could be related to a reduced electron exchange rate.

For CV measurements, the potential values for *j*_pa_ moved within the 0.1–0.2 V window. Again, this could be related to alterations in the electron transfer process and/or to changes in the reference Ag/AgCl electrode, which is quite sensitive to experimental conditions such as the presence of Cl^−^ in PBS, pH 7.4, during AuNS electrodeposition. At the same time, the DPV is characterized by rather stable potential value, changing only slightly in the range of 0.0 to 0.1 V, which is observed due to different nature of the electrochemical signal recording/assessment principles in the CV and DPV techniques.

### 3.3. Electrochemical Characterisation of the Anti-rSpike Detection

The next step of the experiment was to test the ability of the biosensor to detect anti-rSpike. For this purpose, SPCE/AuNS/SAM/rSpike was sequentially incubated with 10 µL of anti-rSpike in a concentration range from 0.5 to 3.5 nM. Each subsequent incubation was accompanied by CV and DPV measurements (Figure 8) in 10 mM PBS, pH 7.4, containing 2 mM [Fe(CN_6_)]^3−/4−^.

CV measurements (Figure 8A) illustrate that stepwise ‘flattening’ of the voltammograms in the anodic region is observed, with a corresponding decrease in *j*_pa_ values, starting from 361.83 ± 0.28 for the solution containing 0 nM of anti-Spike antibodies and decreasing to 270.04 ± 0.63 for the solution with 3.5 nM of anti-Spike antibodies, in the potential window of 0.2 to 0.4 V. The ‘flattening’ of the voltammograms and the potential shifts indicate increasing insulation of the working surface, further hindering access for electrons and changing the value of the redox reaction potential.

DPV experiments revealed the same effect, with a sequential decrease in *j*_p_, i.e., 185.26 ± 1.17, 148.86 ± 1.02, 124.25 ± 0.32, 105.86 ± 0.32, 82.23 ± 0.59, and 66.93 ± 0.2 µA/cm^2^ for 0, 0.5, 1.0, 1.5, 2.5, and 3.5 nM, respectively. In contrast to CV-based experiments, the peaks of the differential pulse voltammograms for solutions with different concentrations of anti-Spike antibodies are characterized by higher resolution and more stable potential values, which correspond to particular concentrations of anti-Spike antibodies.

### 3.4. Limit of Detection and Limit of Quantification

Data gained from the performed electrochemical measurements were used to evaluate the limit of detection (LOD) and limit of quantification (LOQ) for the developed immunosensor, using both the CV and DPV methods. The *j*_pa_ and *j*_p_ values were used as analytical signals for CV and DPV, respectively. Figure 9 shows the calibration curves.

The LOD was calculated as 3.33σ/s and LOQ was calculated as 10σ/s, where σ is the standard deviation for the blank response and s is the slope of the calibration curve [42]. It was revealed that the LOD and LOQ values for the CV-based method were 0.27 nM and 0.81 nM, respectively, while the values calculated from DPV data were 0.14 nM and 0.42 nM, respectively.

### 3.5. Specificity Test

The experiment for nonspecific binding on SPCE/AuNS/SAM/rSpike was performed by comparison of the relative electrochemical signal responses (initial values from Table 2) after incubation of the electrode in 10 mM PBS, pH 7.4, with solutions of 1.5 nM anti-rSpike and 15 nM anti-BSA, (Figure 10). The comparison of the relative responses revealed that for CV, the RR(%) values were 17.80 ± 0.07% and 3.24 ± 0.46% for anti-rSpike and anti-BSA, respectively. Similarly, the RR(%) values for the DPV method were 42.86 ± 0.32% for anti-rSpike and 7.57 ± 0.09% for anti-BSA.

## 4. Conclusions

In this work, electrochemical characterization of SPCE/rSpike and SPCE/AuNS/SAM/rSpike was performed. The electroactive surface area and the heterogeneous electron transfer rate constants were determined and were 22% and 131% higher for SPCE with electrodeposited AuNS, making the SPCE/AuNS surface more suitable for electrochemical measurements. The formation of the SPCE/AuNS/SAM/rSpike biosensing element, as well as the interaction between immobilized rSpike and anti-rSpike, were accompanied by CV and DPV measurements after key stages. For both detection methods, a stepwise decrease in current density was measured after each modification stage, including that applied for the detection of anti-rSpike occurring due to increasingly prohibited access of the [Fe(CN)_6_]^3−/4−^ redox probe to the working electrode. The DPV method was more reliable and more sensitive compared to CV, resulting in 48% lower LOD and LOQ values, making the DPV method more suitable for quantitative analysis. Specificity tests with anti-BSA showed low nonspecific binding for this antibody type. In conclusion, it is expected that the electrochemical immunosensor designed in this research will prove suitable for the diagnosis of the immunological response generated during the course of COVID-19 disease or after vaccination.

## Figures and Tables

**Figure 1 biosensors-12-00593-f001:**
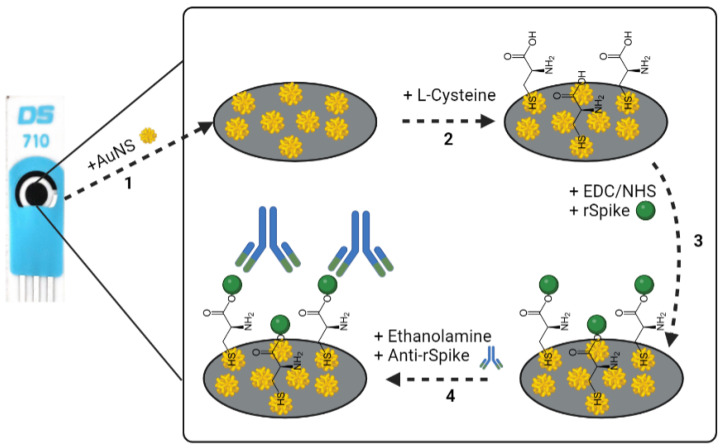
Schematic representation of experimental stages occurring on the SPCE. (**1**): The formation of SPCE/AuNS by electrodeposition; (**2**): SPCE/AuNS/SAM formation; (**3**): the activation of the SPCE/AuNS/SAM by EDC-NHS mixture following SPCE/AuNS/SAM/rSpike formation; (**4**): ethanolamine blocking of remaining active functional groups and SPCE/AuNS/SAM/rSpike/anti-rSpike immunocomplex formation via the interaction between immobilized rSpike protein and the anti-rSpike antibodies present in the aliquot.

**Figure 2 biosensors-12-00593-f002:**
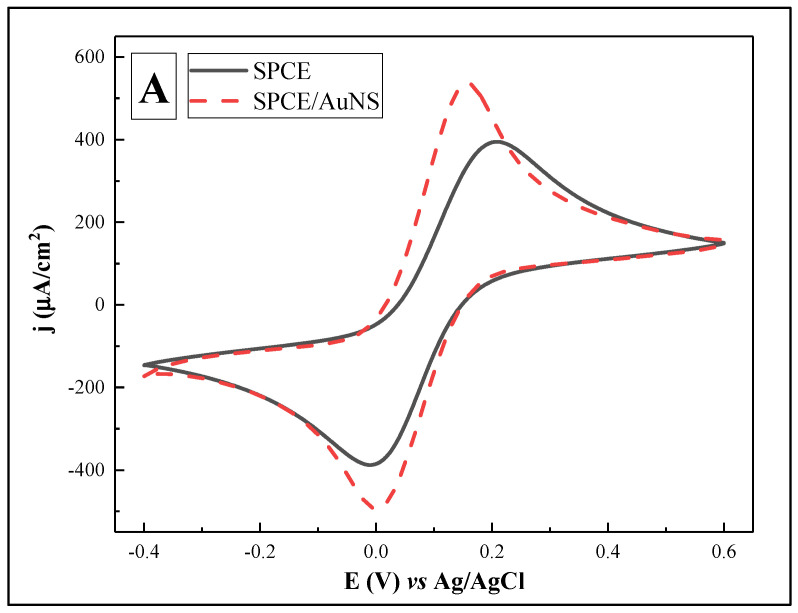
Cyclic voltammograms (**A**) and differential pulse voltammograms (**B**) for SPCE (—) and SPCE/AuNS (- - -). Potential range was from −0.4 to +0.6 V, with a CV scan rate of 0.05 V/s, DPV step size of 0.004 V, pulse height of 0.05 V, pulse period of 100 ms, and pulse width of 50 ms, in 10 mM PBS, pH 7.4, containing 2 mM [Fe(CN_6_)]^3−/4−^. Signal normalized to the geometrical area of the working electrode (0.126 cm^2^).

**Figure 3 biosensors-12-00593-f003:**
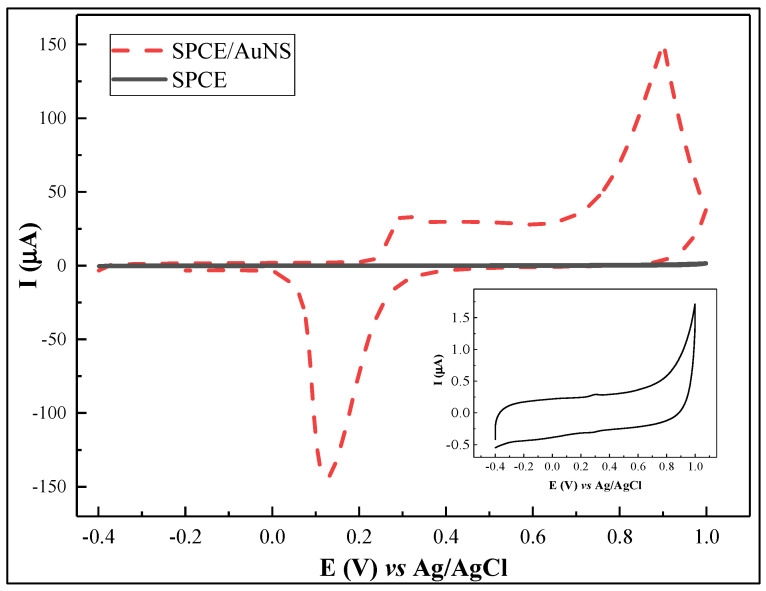
Cyclic voltammogram of SPCE/AuNS in 10 mM H_2_SO_4_. Inset: cyclic voltammogram of SPCE. Potential scan range was from −0.4 to +1.0 V vs. Ag/AgCl, at a scan rate of 0.1 V/s.

**Figure 4 biosensors-12-00593-f004:**
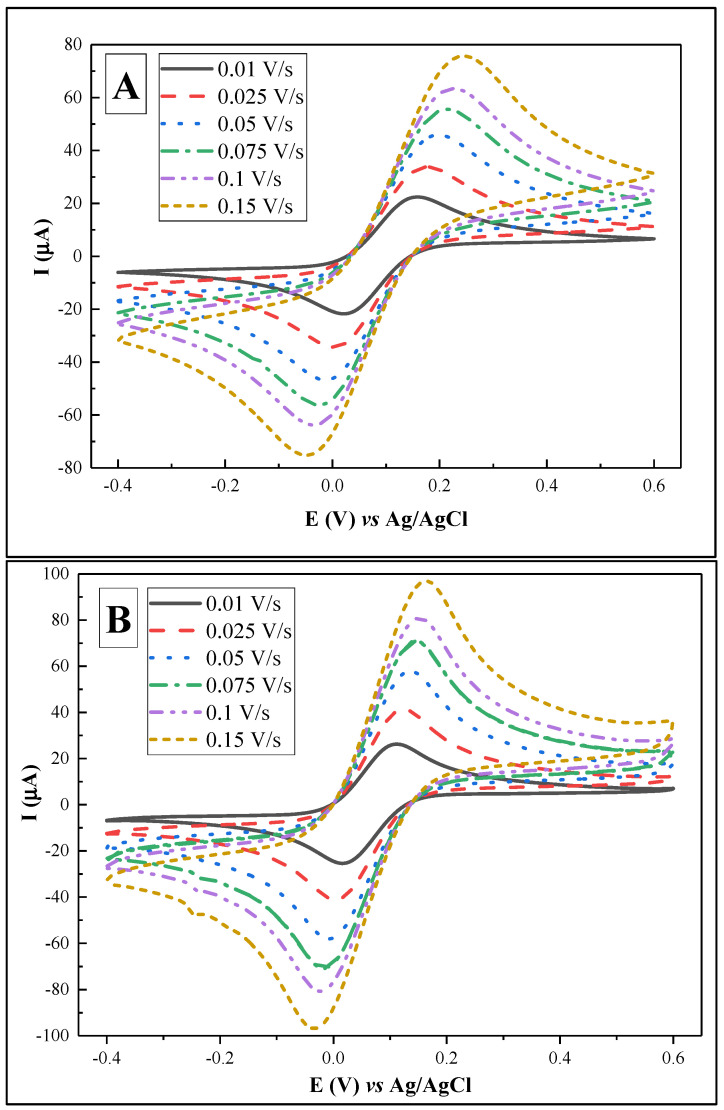
Cyclic voltammograms for SPCE (**A**) and SPCE/AuNS (**B**) at scan rates of 0.01, 0.025, 0.05, 0.075, 0.1, and 0.15 V/s in PBS, pH 7.4, containing 2 mM [Fe(CN_6_)]^3−/4−^.

**Figure 5 biosensors-12-00593-f005:**
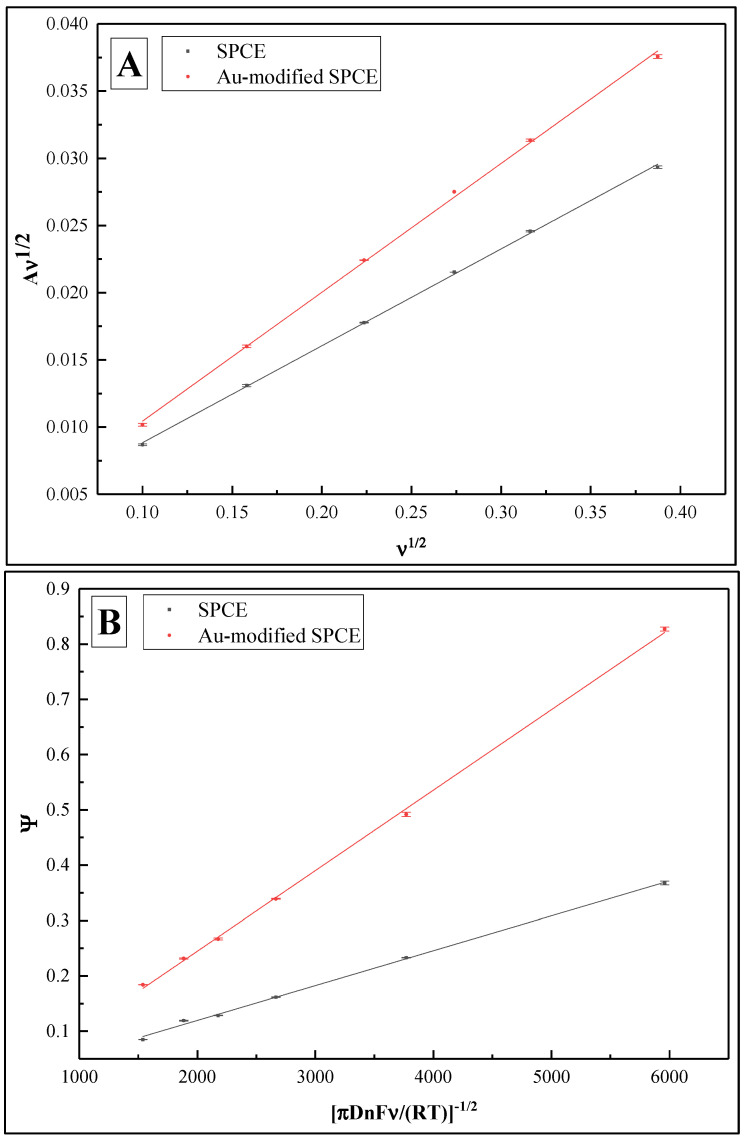
(**A**) Plot of Aν^1/2^ vs. ν^1/2^ showing calculated electrochemically active surface areas of SPCE and SPCE/AuNS as slopes. (**B**) Plot of *Ψ* vs. [π*DnFν*/(*RT*)]^−1/2^ showing calculated *k*^0^ values of SPCE and SPCE/AuNS as slopes.

**Figure 6 biosensors-12-00593-f006:**
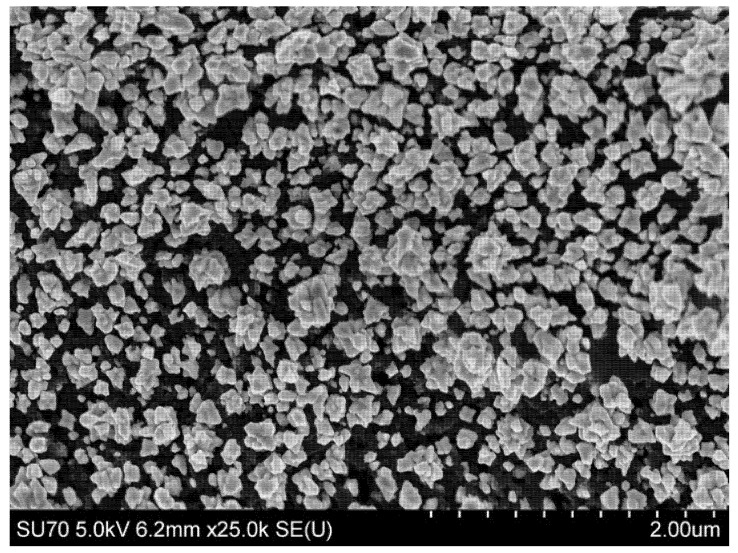
SEM micrograph of SPCE/AuNS.

**Figure 7 biosensors-12-00593-f007:**
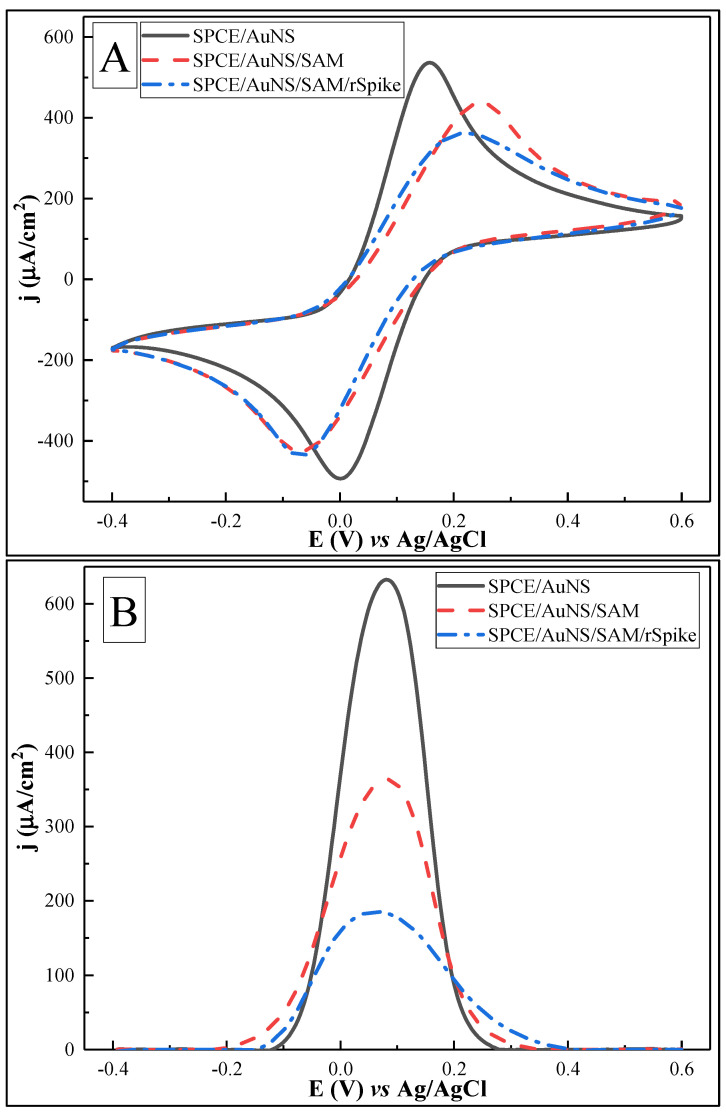
Cyclic voltammograms (**A**) and differential pulse voltammograms (**B**) of SPCE/AuNS (—), after SPCE/AuNS/SAM formation (- - -), and for SPCE/AuNS/SAM/rSpike protein immobilization (-·-). Potential range was from −0.4 to +0.6 V, with a CV scan rate of 0.05 V/s, DPV step size of 0.004 V, pulse height of 0.05 V, pulse period of 100 ms, and pulse width of 50 ms, in 10 mM PBS, pH 7.4, containing 2 mM [Fe(CN_6_)]^3−/4−^. Signal normalized to the geometrical area of the working electrode (0.126 cm^2^).

**Figure 8 biosensors-12-00593-f008:**
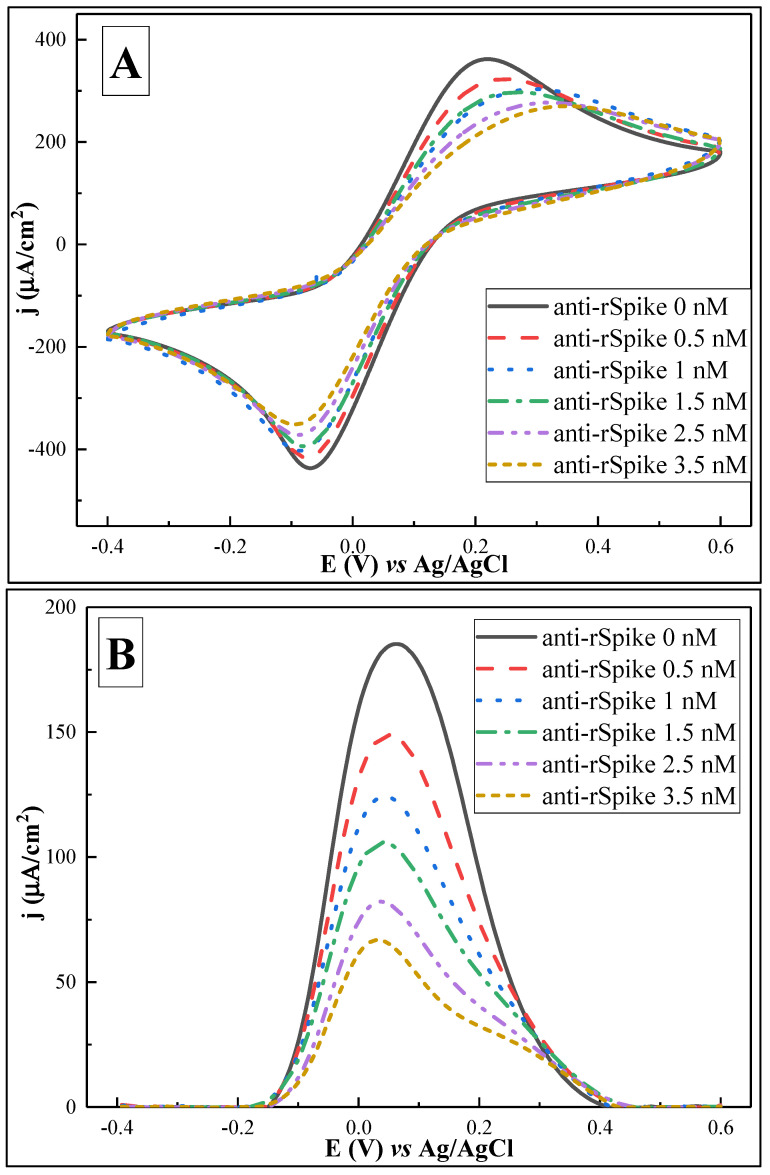
Cyclic voltammograms (**A**) and differential pulse voltammograms (**B**) after interaction with anti-rSpike antibodies of different concentrations (0–3.5 nM). Potential range was from −0.4 to +0.6 V, with a CV scan rate of 0.05 V/s, DPV step size of 0.004 V, pulse height of 0.05 V, pulse period of 100 ms, and pulse width of 50 ms, in 10 mM PBS, pH 7.4, containing 2 mM [Fe(CN_6_)]^3−/4−^. Signal normalised to the geometrical area of the working electrode (0.126 cm^2^). Data are represented as means of three independent experiments.

**Figure 9 biosensors-12-00593-f009:**
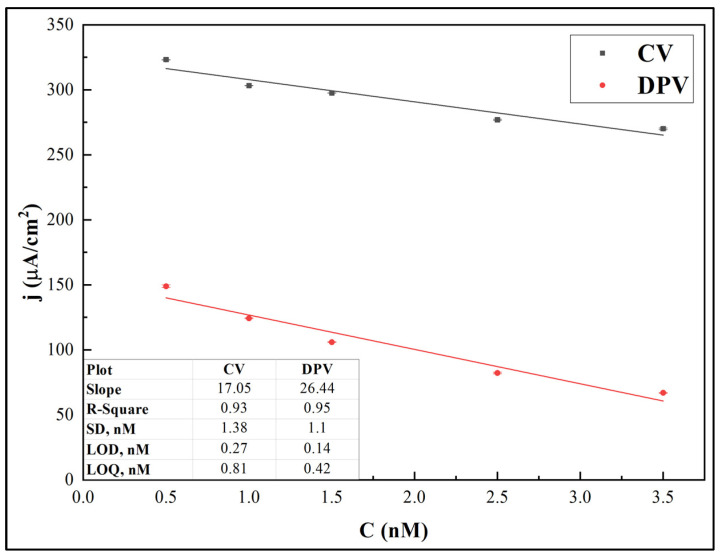
Calibration curves for *j*_pa_ and *j*_p_ obtained from CV and DPV peak values, respectively, vs. anti-rSpike antibody concentration. Error bars are calculated as a percentage of standard error.

**Figure 10 biosensors-12-00593-f010:**
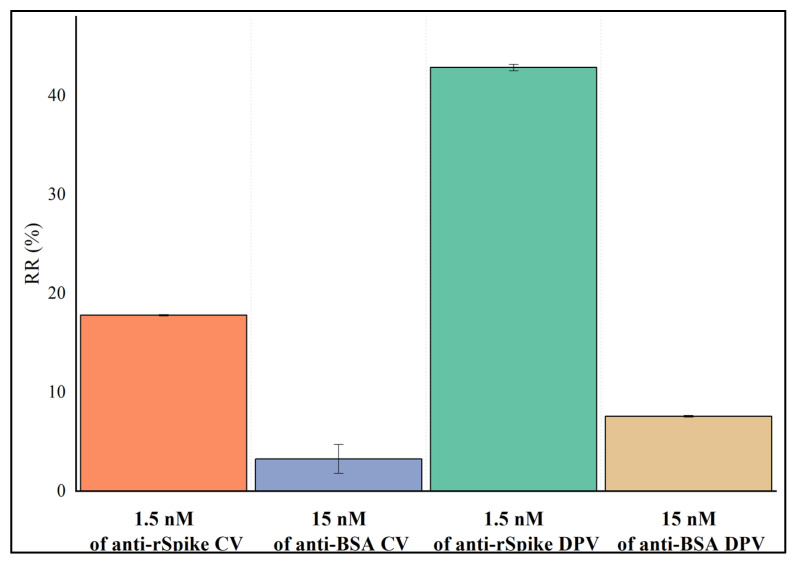
Comparison of the relative responses after 10 min of incubation with 1.5 nM anti-rSpike and 15 nM anti-BSA, for DPV and CV methods. Error bars are calculated as a percentage standard error.

**Table 1 biosensors-12-00593-t001:** Experimental data obtained from CV at different scan rates for SPCE and SPCE/AuNS. Error bars were calculated as a percentage of standard error.

Scan Rate, V/s	SPCE		SPCE/AuNS	
	Ip, A	ΔE, V	Ip, A	ΔE, V
0.01	2.24 × 10^−5^ ± 0.73%	0.14 ± 0.91%	2.63 × 10^−5^ ± 1.05%	0.10 ± 0.44%
0.025	3.38 × 10^−5^ ± 0.65%	0.17 ± 0.15%	4.13 × 10^−5^ ± 0.56%	0.12 ± 0.80%
0.05	4.59 × 10^−5^ ± 0.17%	0.21 ± 0.31%	5.78 × 10^−5^ ± 0.10%	0.14 ± 0.21%
0.075	5.55 × 10^−5^ ± 0.01%	0.24 ± 0.45%	7.10 × 10^−5^ ± 0.00%	0.16 ± 0.66%
0.10	6.34 × 10^−5^ ± 0.20%	0.25 ± 0.66%	8.09 × 10^−5^ ± 0.28%	0.17 ± 0.37%
0.15	7.57 × 10^−5^ ± 0.33%	0.29 ± 0.16%	9.69 × 10^−5^ ± 0.38%	0.20 ± 0.25%

**Table 2 biosensors-12-00593-t002:** Analytical parameters obtained from CV and DPV. Error bars are calculated as a percentage standard error.

	CV	DPV	RR for CV	RR for DPV
	*j*_pa_, µA/cm^2^	*j*_p_, µA/cm^2^	%	%
Au-modified SPCE	536.30 ± 0.42%	632.53 ± 0.83%		
SAM	436.96 ± 0.18%	363.52 ± 0.28%		
rSpike (blank)	361.83 ± 0.28%	185.26 ± 1.17%	0	0
Anti-rSpike 0.5 nM	323.11 ± 0.13%	148.86 ± 1.02%	10.70 ± 0.13	19.65 ± 1.02
Anti-rSpike 1.0 nM	303.18 ± 0.10%	124.25 ± 0.32%	16.21 ± 0.10	32.93 ± 0.32
Anti-rSpike 1.5 nM	297.42 ± 0.07%	105.86 ± 0.32%	17.80 ± 0.07	42.86 ± 0.32
Anti-rSpike 2.5 nM	276.91 ± 0.49%	82.23 ± 0.59%	23.47 ± 0.49	55.61 ± 0.59
Anti-rSpike 3.5 nM	270.04 ± 0.63%	66.93 ± 0.20%	25,370.63	63.87 ± 0.20

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
