# Peer review of "Determination of rSpike Protein by Specific Antibodies with Screen-Printed Carbon Electrode Modified by Electrodeposited Gold Nanostructures"

_biosensors, 2022, doi:10.3390/bios12080593_

Round 1

Reviewer 1 Report

This manuscript reports the detection of rSpike protein by specific antibodies with gold nanostructures modified screen-printed carbon electrode. The experiments are well designed and the results are reliable. The manuscript is well written and of interests to the field. Minor revisions are required before acceptation.

1.     The “physical” in the sentence “Total internal reflection ellipsometry, scanning electrochemical microscopy, surface plasmon resonance, quartz crystal microbalance, electrochemical methods, and other physical methods are among those that can be used to determine analytical signals generated by affinity sensors” is not necessary and could be removed. Because both physical and chemical methods could be used to determine analytical signals generated by affinity sensors. There are many methods could be used to design sensors, so more related references are suggested to be cite, for example 10.1016/j.jobab.2021.06.001; 10.1016/j.aca.2021.339362; 10.1039/c9tb02368b; 10.1016/j.jobab.2021.04.004; 10.1021/acsanm.0c02981; 10.1021/acsami.9b11416; 10.1002/advs.202104066.

2.     “differential pulse DPV” should be written as “DPV”.

3.     For “All experiments were performed at room temperature (20ºC)” and other sentences, there should be a space between the number and unit.

4.     The font size of the numbers in the inset of Figure 3 is too small which is hard to tell.

5.     What does the potential in the Figure 3/4/6/7/8 refer to? The reference electrode should be added in the name of X axial.

Author Response

Invited manuscript for special issue ‘Optical, Electrochemical and Acoustic Methods Based Biosensors for the Investigation of Biomolecules Interactions‘

 ‘Determination of rSpike protein by specific antibodies with screen-printed carbon electrode modified by electrodeposited gold nanostructures’

by: Maryia Drobysh, Viktorija Liustrovaite, Ausra Baradoke, Roman Viter, Chien-Fu Chen, Arunas Ramanavicius, Almira Ramanaviciene

Response to reviewer #1:

We would like to thank the reviewer for the review of our manuscript, valuable comments and recommendations. We did our best in order to improve the manuscript according to revisions recommended by all two reviewers. All the most important changes are highlighted in the revised manuscript.

Reviewer #1 wrote: This manuscript reports the detection of rSpike protein by specific antibodies with gold nanostructures modified screen-printed carbon electrode. The experiments are well designed and the results are reliable. The manuscript is well written and of interests to the field. Minor revisions are required before acceptation.

Response to the reviewer: We will thank the reviewer for positive altitude, comments and recommendations.

Reviewer #1 wrote: The “physical” in the sentence “Total internal reflection ellipsometry, scanning electrochemical microscopy, surface plasmon resonance, quartz crystal microbalance, electrochemical methods, and other physical methods are among those that can be used to determine analytical signals generated by affinity sensors” is not necessary and could be removed. Because both physical and chemical methods could be used to determine analytical signals generated by affinity sensors. There are many methods could be used to design sensors, so more related references are suggested to be cite, for example 10.1016/j.jobab.2021.06.001; 10.1016/j.aca.2021.339362; 10.1039/c9tb02368b; 10.1016/j.jobab.2021.04.004; 10.1021/acsanm.0c02981; 10.1021/acsami.9b11416; 10.1002/advs.202104066.

Response to the reviewer: Corrected according to the comment (‘Introduction’, rows #79-80). Recommended references are cited.

Reviewer #1 wrote: “differential pulse DPV” should be written as “DPV”.

Response to the reviewer: Corrected according to the comment (‘Introduction’, rows #103-104).

Reviewer #1 wrote: For “All experiments were performed at room temperature (20ºC)” and other sentences, there should be a space between the number and unit.

Response to the reviewer: Corrected according to the comment (‘Experimental’, row #139).

Reviewer #1 wrote: The font size of the numbers in the inset of Figure 3 is too small which is hard to tell.

Response to the reviewer: Figure 3 was modified according to the comment.

Reviewer #1 wrote: What does the potential in the Figure 3/4/6/7/8 refer to? The reference electrode should be added in the name of X axial.

Response to the reviewer: Figures 3/4/6/7/8 were modified correspondingly.

Many thanks for the positive feedback.

We thank you for the attention you will pay to this revised version of the manuscript and we sincerely hope that our work after these revisions will be considered as relevant and attractive for publishing.

Yours sincerely,

Arunas Ramanavicius

----------------------------------------------------------------
Prof. habil. dr. Arunas Ramanavicius

Head of Department of Physical Chemistry,

Faculty of Chemistry and Geosciences, Vilnius University, 

Reviewer 2 Report

Drobysh et al. describe a electrochemical immunosensor for the detection of antibodies against SARS-CoV-2 spike proteins utilizing Au nanoparticle deposited on a SPCE by DPV & CV. The protocol is well described and characterized. However, the context and relevance of the work is not clear vs. previous research in this area. The purpose of review for open access is not to judge the perceived importance or originality of the work. However, the motivation for the proposed work and the methodology needs to be explained for the readers to judge the paper. The authors have reported a similar paper utilizing EIS & CV for detection of antibodies against SARS-CoV-2 spike proteins utilizing Au deposited on glass slides. The abstract for Ref.[21] is listed as-"The developed electrochemical immunosensor is suitable for the confirmation of COVID-19 infection or immune response in humans after vaccination" vs. abstract in this current work is "The proposed electrochemical immunosensor is suitable for confirming COVID-19 infection or immunological response in human beings after the vaccination." The antibodies are also the same, collected in same manner. The differences are in the type of electrode (SPCE vs. Au on glass) and DPV vs. EIS. The authors need to add context on the relevance of current work vs. ref [21]. For instance, why should readers be interested in DPV vs. EIS, pros/cons between two of the proposed protocols (say SPCE vs. Au electrodes). References to similar immunosensors for COVID-19 antibodies should also be published for giving context to readers (even if the LOD/LOQ might be lower or higher). For instance, other research into EC sensors for COVID-19 antibodies or active infection. Also very briefly cite/mention any previous papers or reviews which have used similar systems (say Au or similar nanoparticles for EC immunosensors).

Few other aspects-
1) Please provide statistical relevance for the data. For instance please mention number of replicates for all the data that is listed. For instance, CV curves are n=1 for Fig. 3, 4, 5, 6 (except maybe Fig. 9). If mean values for n=3 replicates are listed please explicitly state the same (for say Table 2). This should be listed for readers to understand statistical significance of data. How many SPCEs were used for these data points, is there electrode to electrode variability which affects the LODs etc.

2) Based on our previous work [21], it was assumed that electron charge transfer in the cathodic region is slower than in the anodic region ->What is the mechanism for this if known? Ref.[21] uses a different type of electrode so is the results relevant the different variables between the two? Please explain more.

3) Please provide catalog number for reagents or instruments when available. Please provide DropSens catalog numbers e,g. for SPCEs & box connector (BIDSC-FET?).

4) Please provide scan rate for DPV.

5) "was 5860 BAU/mL" - Is this from a ref. or was this done as part of this expt. Please explain source of this value (or if this is from ref.[22-24]?).

Author Response

Invited manuscript for special issue ‘Optical, Electrochemical and Acoustic Methods Based Biosensors for the Investigation of Biomolecules Interactions‘

 ‘Determination of rSpike protein by specific antibodies with screen-printed carbon electrode modified by electrodeposited gold nanostructures’

by: Maryia Drobysh, Viktorija Liustrovaite, Ausra Baradoke, Roman Viter, Chien-Fu Chen, Arunas Ramanavicius, Almira Ramanaviciene

Response to reviewer #2:

We would like to thank the reviewer for the review of our manuscript, valuable comments and recommendations. We did our best in order to improve the manuscript according to revisions recommended by all two reviewers. All the most important changes are highlighted in the revised manuscript.

Reviewer #2 wrote: Drobysh et al. describe a electrochemical immunosensor for the detection of antibodies against SARS-CoV-2 spike proteins utilizing Au nanoparticle deposited on a SPCE by DPV & CV. The protocol is well described and characterized. However, the context and relevance of the work is not clear vs. previous research in this area. The purpose of review for open access is not to judge the perceived importance or originality of the work. However, the motivation for the proposed work and the methodology needs to be explained for the readers to judge the paper. The authors have reported a similar paper utilizing EIS & CV for detection of antibodies against SARS-CoV-2 spike proteins utilizing Au deposited on glass slides. The abstract for Ref.[21] is listed as-"The developed electrochemical immunosensor is suitable for the confirmation of COVID-19 infection or immune response in humans after vaccination" vs. abstract in this current work is "The proposed electrochemical immunosensor is suitable for confirming COVID-19 infection or immunological response in human beings after the vaccination." The antibodies are also the same, collected in same manner. The differences are in the type of electrode (SPCE vs. Au on glass) and DPV vs. EIS.

Response to the reviewer: We will thank the reviewer for positive altitude, comments and recommendations.

Reviewer #2 wrote: The authors need to add context on the relevance of current work vs. ref [21]. For instance, why should readers be interested in DPV vs. EIS, pros/cons between two of the proposed protocols (say SPCE vs. Au electrodes). References to similar immunosensors for COVID-19 antibodies should also be published for giving context to readers (even if the LOD/LOQ might be lower or higher). For instance, other research into EC sensors for COVID-19 antibodies or active infection. Also very briefly cite/mention any previous papers or reviews which have used similar systems (say Au or similar nanoparticles for EC immunosensors).

Response to the reviewer: Corresponding information was added to ‘Introduction’ (rows #63-66; 83-98).

Reviewer #2 wrote: Please provide statistical relevance for the data. For instance please mention number of replicates for all the data that is listed. For instance, CV curves are n=1 for Fig. 3, 4, 5, 6 (except maybe Fig. 9). If mean values for n=3 replicates are listed please explicitly state the same (for say Table 2). This should be listed for readers to understand statistical significance of data. How many SPCEs were used for these data points, is there electrode to electrode variability which affects the LODs etc.

Response to the reviewer: Corresponding information was added to the descriptions of Figures 3-8 and Table 2.

Reviewer #2 wrote: Based on our previous work [21], it was assumed that electron charge transfer in the cathodic region is slower than in the anodic region ->What is the mechanism for this if known? Ref.[21] uses a different type of electrode so is the results relevant the different variables between the two? Please explain more.

Response to the reviewer: Corresponding information was added to ‘Results and discussion’ (rows #237-243).

Reviewer #2 wrote: Please provide catalog number for reagents or instruments when available. Please provide DropSens catalog numbers e,g. for SPCEs & box connector (BIDSC-FET?).

Response to the reviewer: Corresponding information was added to ‘Experimental’ (rows #130, 133).

Reviewer #2 wrote: Please provide scan rate for DPV.

Response to the reviewer: All adjustable parameters of the used method were specified in the descriptions of the corresponding graphs.

Reviewer #2 wrote: "was 5860 BAU/mL" - Is this from a ref. or was this done as part of this expt. Please explain source of this value (or if this is from ref.[22-24]?).

Response to the reviewer: Corresponding information was added to ‘Experimental’ (rows #169-171).

Many thanks for the positive feedback.

We thank you for the attention you will pay to this revised version of the manuscript and we sincerely hope that our work after these revisions will be considered as relevant and attractive for publishing.

Yours sincerely,

Arunas Ramanavicius

----------------------------------------------------------------
Prof. habil. dr. Arunas Ramanavicius

Head of Department of Physical Chemistry,

Faculty of Chemistry and Geosciences, Vilnius University, 

Round 2

Reviewer 2 Report

The changes based on relevant review comments have been incorporated.